# Graph Contrastive Learning via Weisfeiler-Leman Dual-View Sampling

## Abstract

Graph contrastive learning (GCL) approaches have gained momentum over the past few years. By augmenting the original graph data, the common GCL pipeline learns from such multiple contrastive graph views in a self-supervised manner, tackling critical issues in the literature, such as node label scarcity. To obtain contrastive views, most GCL techniques heavily rely on feature-space similarity measures. We consider this as a limiting factor in GCL, since it implies that node features are (in general) informative and closely aligned with the graph topology, an assumption that does not hold, for instance, in the case of heterophilic graphs. In this work, we propose to address the problem by coupling the usual feature-space similarities with structure-based measures, which we propose to implement through the Weisfeiler-Leman (WL) family of algorithms. Our framework, dubbed WLGCL, introduces a dual-view sampling strategy that works on features- and WL-level to construct more reliable contrastive pairs. WLGCL integrates a multi-positive and hard-negative contrastive loss to ensure the alignment-uniformity trade-off without modifying the design of the graph encoder. Extensive experiments on six benchmark datasets and against seven state-of-the-art baselines demonstrate the efficacy of WLGCL, where additional empirical evaluations justify the adoption of our architectural choices for the model.

## 1 Introduction

Self-supervised learning (SSL) (Gui et al., 2024) has emerged as a powerful paradigm for learning generalizable representations without human-annotated labels. By designing pretext tasks that exploit the inherent structure of data, SSL enables models to extract semantic patterns that transfer effectively to downstream tasks (Tomasev et al., 2022). Among various SSL approaches, contrastive learning (CL) has become one of the most influential frameworks. CL learns discriminative embeddings by encouraging representations of semantically related samples to be similar while enforcing separation between unrelated ones. Originally developed in computer vision with methods such as SimCLR (Chen et al., 2020) and MoCo (He et al., 2020), CL has since been extended to images (Radford et al., 2021), text (Gao et al., 2021), and audio (Baevski et al., 2020), demonstrating that well-designed positive-negative relational constraints can drive robust feature learning across diverse modalities.

Motivated by this success, recent years have witnessed rapid progress in graph contrastive learning (GCL) (Velickovic et al., 2019; You et al., 2020; Zhu et al., 2020), which adapts contrastive principles to relational and non-Euclidean data. GCL methods typically use paired graph augmentations or cross-view transformations to enforce invariance, encouraging the model to capture both topological structure and attribute information. A wide spectrum of design choices (*e.g.*, view generation, contrastive objectives, sampling strategies, and structural granularity) has led to a variety of methods, including GRACE (Zhu et al., 2020) and BGRL (Thakoor et al., 2022), as well as spectral and subgraph-based approaches such as PolyGCL (Chen et al., 2024) and FOSSIL (Sangare et al., 2025). These techniques have achieved strong performance in node classification, clustering, and link prediction, underscoring the versatility of contrastive objectives in graph representation learning.

Despite these advancements, most GCL frameworks still rely heavily on feature-space similarity to construct positive and negative samples. Whether the sampling is explicit or implicit, similarity is determined almost exclusively by distances in the learned embedding space. This assumption works well when node features are informative and strongly aligned with graph topology, as in homophilic graphs. However, it often breaks down in heterophilic or structurally complex settings, where nodes sharing similar structural roles may look entirely different in feature space (Liu et al., 2023). Consequently, feature-based sampling may generate (i) false positives, *i.e.,* nodes that appear close in embeddings but are structurally unrelated; and (ii) weak negatives, *i.e.*, structurally similar nodes that are mistakenly pushed apart. These issues are especially problematic in graph contrastive learning because positive-pair construction directly determines the geometry of the learned representation space. When structurally inconsistent positives are enforced during training, the learned embeddings may overfit superficial feature correlations while failing to preserve meaningful topological roles.

This motivates the need for a structure-aware contrastive sampling strategy capable of explicitly modeling structural consistency independently of feature similarity. A desirable structural similarity measure for graph contrastive learning should satisfy several properties: (i) sensitivity to higher-order neighborhood structure beyond local degree statistics, (ii) permutation invariance, and (iii) scalability to large graphs. Simple heuristics such as node degree or common-neighbor counts are often insufficient, as they fail to distinguish nodes with similar local statistics but fundamentally different multi-hop structural contexts. Meanwhile, the Weisfeiler-Leman (WL) family of algorithms offers a principled and expressive way to capture multi-scale structural patterns (Weisfeiler & Leman, 1968). WL refinements, widely used in graph kernels and GNN expressivity analysis, provide structure-aware node similarity that is invariant to permutations and independent of feature noise (Morris et al., 2019). However, despite their strong structural discrimination capabilities, WL-based similarities remain largely underutilized in contrastive learning, which continues to rely predominantly on feature-driven heuristics.

To bridge this gap, we propose WLGCL, a dual-view sampling framework that integrates feature-space similarity with WL-based structural similarity to construct more reliable contrastive pairs. Rather than assuming that embeddings implicitly encode structure, we decouple the two notions of similarity and treat their intersection as trustworthy positives, while using the mismatched cases as naturally informative hard negatives. More fundamentally, our approach reframes contrastive sampling as a consistency problem across views, where agreement between feature-based and structural similarities serves as a signal of semantic reliability. This perspective provides a principled alternative to conventional feature-driven sampling and clarifies the role of structure in contrastive learning. Building upon this sampling strategy, we further introduce a multi-positive contrastive objective that enhances the alignment-uniformity trade-off (Wang et al., 2024; Yang et al., 2023; Yan et al., 2024; Zhu & Qiu, 2026) without altering the encoder design. Across six benchmark datasets spanning homophilic and heterophilic settings, WLGCL consistently outperforms state-of-the-art baselines while maintaining competitive training efficiency.

## 2 Related Work

This work is related to two lines of research: graph contrastive learning (GCL) and the Weisfeiler-Leman (WL) algorithm. GCL provides a self-supervised framework for learning graph representations by contrasting multiple views, while the WL algorithm offers a principled way to capture graph structural equivalence. Our method connects these two directions by incorporating WL-based structural similarity into contrastive pair construction. We review GCL first, followed by the WL algorithm.

### 2.1 Self-Supervised and Contrastive Learning on Graphs

Contrastive learning (CL) became popular for the first time in computer vision (Radford et al., 2021), where methods such as SimCLR (Chen et al., 2020) and MoCo (He et al., 2020) demonstrated that meaningful representations can be learned without manual labels by contrasting positive and negative pairs. This paradigm has since been successfully extended to other modalities, such as text (Gao et al., 2021), and audio (Baevski et al., 2020), and more recently has been applied to graph-structured data, giving rise to graph contrastive learning (GCL). The common self-supervised GCL framework learns node- or graph-level

representations by contrasting multiple correlated views of a graph (Velickovic et al., 2019; You et al., 2020; Zhu et al., 2020). By enforcing invariance across views while maintaining embedding uniformity, GCL has shown strong performance on downstream tasks, including node classification, clustering, and link prediction (Liu et al., 2023). The contrastive objectives have further been refined through improved sampling and optimization strategies, such as hard-negative mining in MoCHi (Kalantidis et al., 2020). As we will show, we extend these works to a structure-aware, multi-positive setting on graphs.

A typical GCL pipeline consists of three components: a graph encoder, a view generation module, and a contrastive objective. The encoder is usually implemented using message-passing GNNs such as GCN (Kipf & Welling, 2017), GAT (Velickovic et al., 2018), or GIN (Xu et al., 2019). The view generator constructs multiple correlated graph representations via stochastic perturbations, including feature masking, edge or node dropping, subgraph sampling, diffusion-based propagation, or spectral filtering. The contrastive objective then encourages representations of the same instance across views to be similar, while separating representations of different instances, commonly through InfoNCE-style losses (van den Oord et al., 2018) or mutual information maximization.

Early GCL methods primarily focused on node-to-node or graph-to-graph contrast based on augmented views. GRACE (Zhu et al., 2020), for instance, relies on stochastic feature and edge perturbations to generate paired views, while DGI (Velickovic et al., 2019) maximizes mutual information between local node embeddings and a global summary representation. However, such approaches are often sensitive to augmentation design and negative-sample bias. To improve training stability, augmentation-free or negative-free methods have been proposed, moving beyond purely contrastive objectives. MUSE (Yuan et al., 2023) combines contrastive alignment with semantic masking and reconstruction to better handle heterophilic graphs. Beyond contrastive techniques, BGRL (Thakoor et al., 2022) adopts a bootstrap self-distillation strategy with a momentum-updated target encoder.

More recent approaches explore richer structural and spectral perspectives in GCL. PolyGCL (Chen et al., 2024) introduces learnable polynomial filters to generate frequency-aware views, while EPAGCL (Xu et al., 2025) provides theoretical analysis linking contrastive objectives with spectral smoothing effects. SpeGCL (Shou et al., 2025) further departs from explicit positive-pair construction by enforcing spectral consistency without predefined positives. In parallel, AutoGCL (Yin et al., 2022) automates view generation through learnable augmentation policies, while subgraph-level methods such as FOSSIL (Sangare et al., 2025) leverage optimal transport to align higher-order structural patterns across sampled subgraphs.

Despite these advances, most existing GCL methods still define contrastive pairs predominantly based on feature-space similarity, implicitly assuming that learned embeddings faithfully encode structural roles. This assumption often fails on graphs where feature similarity and structural similarity diverge, motivating the need for more reliable, structure-aware sampling strategies, which we address in this work.

## 2.2 Weisfeiler-Leman Algorithm

The Weisfeiler-Leman (WL) algorithm is a classical heuristic for graph isomorphism testing that iteratively refines node representations based on local neighborhood structure (Weisfeiler & Leman, 1968). Starting from an initial node labeling, WL repeatedly aggregates the labels of neighboring nodes to produce increasingly fine-grained partitions of the vertex set. Through this refinement process, nodes that share similar structural roles tend to receive identical colors, while structurally distinct nodes are progressively separated. Although the WL algorithm cannot distinguish all non-isomorphic graphs (for instance, certain regular graphs remain indistinguishable under standard WL refinement (Kriege et al., 2018)), it is highly effective for a broad class of graphs and has been widely adopted as a principled notion of structural similarity (Babai & Kucera, 1979). Its ability to capture multi-scale topological patterns makes it particularly useful for comparing nodes and graphs in a permutation-invariant manner. At each iteration, a node updates its representation according to the multiset of neighboring node labels, progressively capturing higher-order topological context beyond immediate connectivity patterns. Consequently, nodes that remain indistinguishable across multiple WL refinement iterations tend to share similar multi-hop structural contexts and functional roles within the graph, even when their raw features differ substantially (Babai & Kucera, 1979). Importantly, WL refinement captures substantially richer structural information than simple heuristics such as node degree or

common-neighbor statistics. While local measures only characterize immediate connectivity, WL iteratively aggregates neighborhood compositions across multiple hops, enabling discrimination between nodes with identical local statistics but different higher-order structural contexts.

WL-based refinements have played a central role in graph learning (Morris et al., 2023). Early work, such as the WL-subtree kernel, demonstrated strong performance in graph classification by explicitly exploiting the refinement hierarchy (Shervashidze et al., 2011). More recently, it has been shown that message passing graph neural networks (GNNs) follow a computation pattern analogous to the WL algorithm, iteratively updating node embeddings through neighborhood aggregation (Xu et al., 2019; Morris et al., 2019). As a result, the expressive power of standard GNNs is upper-bounded by the discriminative ability of the WL test.

Extensions of the WL algorithm to higher dimensions further enhance its expressive capacity by refining tuples of nodes rather than individual vertices (Cai et al., 1992). These higher-order WL variants have inspired the design of more powerful graph neural architectures and are commonly used as theoretical tools for analyzing GNN expressiveness (Morris et al., 2020). Overall, the WL family of algorithms provides a principled and structure-aware foundation for graph representation learning, motivating its still-unexplored integration into modern self-supervised and contrastive learning frameworks.

## 3 Proposed Methodology

In graph contrastive learning (GCL), positive and negative samples are typically constructed based on feature similarity in the embedding space. This implicitly assumes that learned representations faithfully capture the underlying graph structure, an assumption that may not hold in many real-world graphs where feature similarity and structural roles are misaligned. As a result, feature-based sampling often introduces false positives, nodes that appear close in the embedding space yet are structurally unrelated, as well as weak negatives, since randomly or queue-based sampled negatives may include nodes that are structurally or functionally similar to the anchor. This motivates the need for a more robust sampling strategy that explicitly incorporates structural information alongside feature-based similarity.

To this end, we propose WLGCL, a dual-view sampling framework that integrates feature similarity and WL-based structural similarity for contrastive pair construction. Our approach is grounded in the expressivity of the WL algorithm. It generates a hierarchical WL tree capturing multi-hop structural patterns for each node and serves as the structural backbone of our framework. Figure 1 gives a example of this WL iterations. Based on this, WLGCL proceeds in two stages: (i) dual-view candidate retrieval, and (ii) intersection-based positive extension with hard negative mining. In stage (i), for each anchor node, we independently retrieve candidate neighbors from the feature embedding space and the WL structural space, forming two complementary views of similarity. In stage (ii), we take the intersection of these two sets as reliable positives, while nodes appearing in only one view are treated as informative hard negatives. This design enables the model to retain semantically consistent positives while explicitly identifying samples that are misleading from either perspective, overcoming limitations of purely feature-based or random sampling strategies. Finally, the learned representations are optimized using a contrastive loss, where our sampling module acts as a plug-and-play component without modifying the encoder architecture or objective function.

In the following, we deepen into each component of the framework. Before describing them, we also provide some background notions and notation regarding graph learning methods. A visual representation of the complete WLGCL pipeline is provided in Figure 2.

### 3.1 Background on Graph Learning

Without loss of generality, in this work, we consider undirected graphs with node features. Let $\mathcal{G} = (\mathcal{V}, \mathcal{E})$ be an undirected graph, where $v \in \mathcal{V}$ and $(v, u) \in \mathcal{E}$ represent a node and an edge within the graph, respectively. We indicate with $\mathbf{X} \in \mathbb{R}^{|\mathcal{V}| \times F}$ the vector of original node features, where $\mathbf{x}_v \in \mathbb{R}^F$ stands for the features of node $v \in \mathcal{V}$. Then, we use $\mathcal{P}_v$ to represent the set of one-hop neighbors of node $v$, such that $\mathcal{P}_v = \{p \in \mathcal{V} : (v, p) \in \mathcal{E}\}$. Following the typical graph contrastive learning (GCL) terminology, $\mathcal{P}_v$ is also defined as the set of *positive* nodes for the anchor node $v$, as opposed to its *negative* nodes

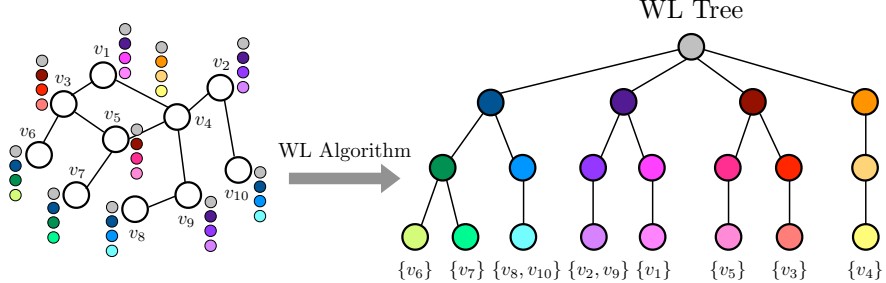

Figure 1: WL algorithm example: 1) On the left side is a graph $\mathcal{G}$ with uniform initial colors and refined colors (ordered from top to bottom) with iteration $t = 3$; 2) On the right side is the corresponding WL tree of the graph.

$\mathcal{N}_v = \{n \in \mathcal{V} \colon (v, n) \notin \mathcal{E}\}$. Apart from these general definitions, in the following sections, we will introduce specific cases of positive and negatives samples for node $v$. In common graph learning tasks (such as node classification or link prediction), we seek to learn an encoding function over $\mathcal{G}$ to perform the downstream task (such as classifying node labels or predicting the existence of an edge between two nodes). For instance, message passing graph neural networks (GNNs) constitute powerful encoders that map $\mathcal{G}$ to latent node features $\mathbf{H} \in \mathbb{R}^{|\mathcal{V}| \times D}$ such that $\mathbf{H} = \text{GNN}(\mathcal{G}, \mathbf{X})$, where $\mathbf{h}_v \in \mathbb{R}^D$ is the node embedding updated after the message passing with the GNN.

## 3.2 WL Tree Refinement Process

The goal of the Weisfeiler-Leman (WL) part in WLGCL is to explicitly recover structural consistency that may not be reflected in feature-space similarity. In many graphs, nodes occupying similar topological roles can exhibit substantially different attributes, making feature-only sampling unreliable. WL refinement addresses this issue by encoding multi-hop rooted subtree structures, allowing structurally equivalent nodes to be identified independently of their raw features or learned embeddings. We employ the WL refinement procedure to derive structure-aware node similarities used in our dual-view contrastive sampling framework. For each node $v \in \mathcal{V}$, we initialize it with a color $C_v^{(0)}$, which can be derived from node attributes or assigned uniformly when no features are available. The WL algorithm iteratively refines node colors by aggregating the multiset of colors of neighboring nodes.

Formally, the coloring at iteration $t$ partitions the set of nodes $\mathcal{V}$ into color classes, in the sense that nodes assigned the same color belong to the same class. The new color of a node $v$ at iteration $t$ is given by:

$$C_v^{(t)} = (C_v^{(t-1)}, \{\!\!\{C_u^{(t-1)} \colon u \in \mathcal{P}_v\}\!\!\}) \quad t \in \{1, 2, \ldots, T\}. \tag{1}$$

Hence, the new color of $v$ depends on its previous color and on the multiset of the colors of its neighbors in the previous iteration. It is then clear that in iteration $t$, two vertices $v$ and $u$ receive different colors if they already had different colors in iteration $t-1$ or if the multisets of their neighbors' colors at iteration $t-1$ differ. Since the implication $C_v^{(t-1)} \neq C_u^{(t-1)} \Rightarrow C_v^{(t)} \neq C_u^{(t)}$ holds for all $v, u \in \mathcal{V}$ and any $t \in \mathbb{N}$, the color classes produced in iteration $t$ are at least as fine as those produced in iteration $t-1$. That is, if two nodes belonged to different color classes in iteration $t-1$ of the algorithm, they will surely belong to different classes in the next iteration. Conversely, if they belonged to the same color class in iteration $t-1$, they may or may not be assigned to the same class in iteration $t$. Hence, the sequence of colorings produced across WL iterations induces a family of nested subsets, which can naturally be represented by a hierarchy.

Figure 1 illustrates a graph together with the colors produced by three iterations of the WL algorithm (from top to bottom). It also shows the hierarchy (WL tree) induced by the refinement process. After three iterations, there are eight color classes, each containing either a single node or two nodes of the input graph. Nodes that remained in the same color class across multiple WL iterations are more structurally similar than nodes that never shared a color class or remained in the same color class across less WL iterations. We adopt WL refinement because it simultaneously provides (i) higher-order structural expressivity, (ii)

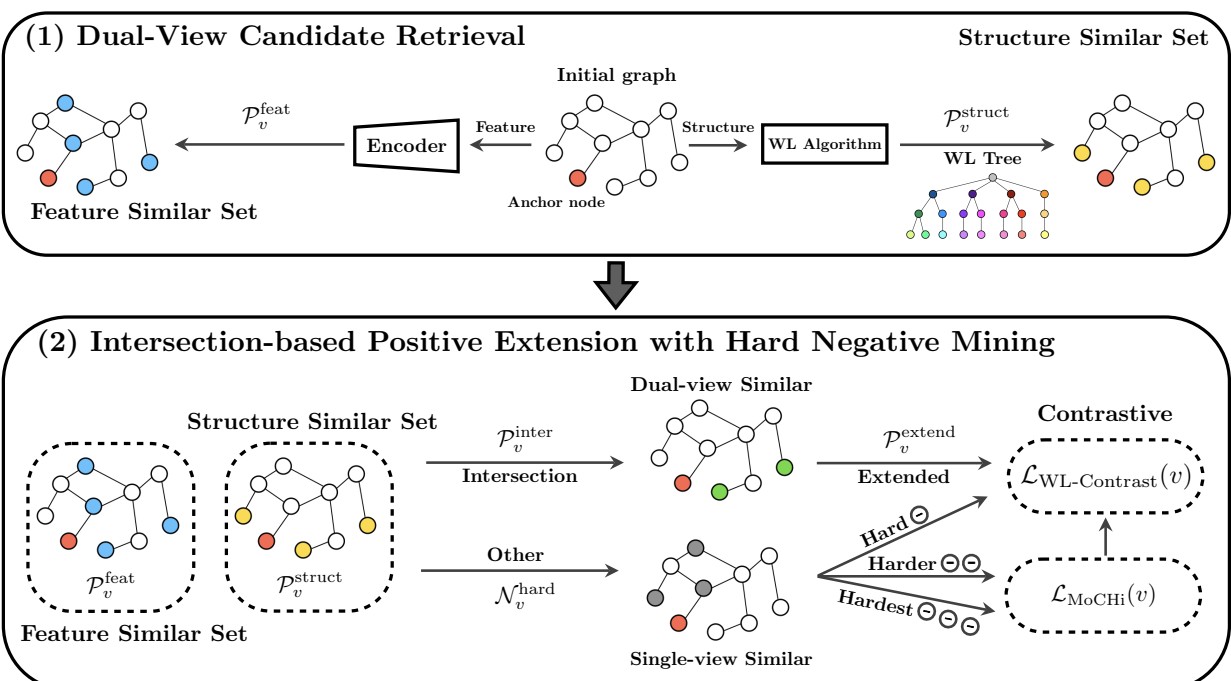

Figure 2: Overview of WLGCL, consisting of two main stages. In the first stage (above), WLGCL retrieves candidate neighbors from the feature embedding and WL structural spaces (left and right, respectively). Then, in the second stage (below), WLGCL computes the intersection of the feature and structure similar sets to obtain a reliable set of positive nodes, taking nodes appearing in only one of the two sets as hard negatives; all such node sets are used to optimize the model's contrastive loss function, involving also a MoCHi-based (Kalantidis et al., 2020) loss component requiring additional levels of harder and hardest negative node samples.

permutation invariance, and (iii) computational scalability. These properties make WL particularly suitable for contrastive sampling, where structural similarity must be both discriminative and efficiently computable over large graphs.

### 3.3 Dual-View Candidate Retrieval

Given an anchor node $v$, the first stage of our framework independently retrieves two candidate sets of semantically related nodes: one based on feature similarity in the embedding space and the other based on structural similarity in the original graph topology. This separation is crucial, as the former reflects what the encoder currently assumes to be similar, while the latter reflects intrinsic structural patterns that are independent of representation learning.

**Feature-based similarity selection.** First, we encode the input graph $\mathcal{G}$ along its node features through a GNN encoder to obtain node embeddings $\mathbf{H} = \mathrm{GNN}(\mathcal{G}, \mathbf{X})$. While we maintain that our framework is technically encoder-agnostic and could be seamlessly combined with other GNN backbones, we adopt WLHN as the default encoder in our implementation due to its strong capability in capturing higher-order structural patterns and preserving multi-scale neighborhood information.

For each anchor node (marked **red** 🔴 in Figure 2), we compute the cosine similarity to all other nodes:

$$S_{\text{feat}}(v, u) = \cos(\mathbf{h}_v, \mathbf{h}_u) = \frac{\mathbf{h}_v^\top \mathbf{h}_u}{\|\mathbf{h}_v\| \cdot \|\mathbf{h}_u\|}. \tag{2}$$

On such a basis, we establish a pre-defined threshold $\theta_{\text{feat}}$ according to which nodes with feature-based similarity $S_{\text{feat}}$ above it will belong to the feature-consistent candidate set (marked **blue** 🔵 in Figure 2):

$$\mathcal{P}_v^{\text{feat}} = \{u \colon S_{\text{feat}}(v, u) > \theta_{\text{feat}}\}, \quad \forall v \in \mathcal{V}. \tag{3}$$

In the equation above, $\mathcal{P}_v^{\text{feat}}$ indicates the set of nodes whose GNN encoded features are similar under a certain threshold $\theta_{\text{feat}}$. Setting a suitable threshold value allows a finer selection of the most similar node embeddings to the anchor one.

**Structure-based similarity selection.** At the same time, we measure the structural similarity using a Weisfeiler-Leman (WL) refinement process. Given a graph $\mathcal{G}$, we perform $T$ iterations of WL refinement and construct a WL tree, where each node $v \in \mathcal{V}$ is mapped to a unique leaf node after $T$ iterations. Each leaf corresponds to the final refined WL label that encodes the rooted $T$-hop neighborhood structure of the node.

We define the WL-tree distance between two nodes $v$ and $u$ as the shortest path length between their corresponding leaf nodes in the WL tree:

$$S_{\text{struct}}(v, u) = \text{sp}(C_v^{(T)}, C_u^{(T)}), \tag{4}$$

where $\text{sp}(\cdot)$ is the function computing the shortest path distance between the two leaf nodes $v$ and $u$ within the $T$-iterations WL-tree. A smaller distance indicates higher structural similarity, as the two nodes share more common refinement history, while a larger distance implies greater structural dissimilarity.

Based on this distance, and similarly to the feature-based similarity case, we retain the most similar candidate nodes according to the structural similarity (marked in **yellow** 🟡 in Figure 2) as:

$$\mathcal{P}_v^{\text{struct}} = \{u \colon S_{\text{struct}}(v, u) < \theta_{\text{struct}}\} \quad \forall v \in \mathcal{V}, \tag{5}$$

where $\theta_{\text{struct}} > 0$ is a predefined threshold controlling the maximum admissable WL-tree distance. In practice, $\theta_{\text{struct}}$ is bounded by the maximum depth of the WL tree (twice of the maximum depth) and is selected as a hyperparameter based on validation performance. Since this criterion depends solely on the graph topology encoded by WL refinement, the resulting candidate set is encoder-agnostic.

### 3.4 Intersection-Based Positive Extension and Hard Negative Mining

In standard GCL, each anchor node in one augmentation view has a single positive, namely, its counterpart in the other view that corresponds to the same underlying node. Negatives are drawn from all other nodes. In contrast, WLGCL extends the positive set by exploiting the dual-view consistency discovered during the candidate retrieval described above.

Given the feature-consistent candidate set $\mathcal{P}_v^{\text{feat}}$ of (3) and the structure-consistent candidate set $\mathcal{P}_v^{\text{struct}}$ of (5), we first compute their intersection:

$$\mathcal{P}_v^{\text{inter}} = \mathcal{P}_v^{\text{feat}} \cap \mathcal{P}_v^{\text{struct}} \quad \forall v \in \mathcal{V}, \tag{6}$$

where $\mathcal{P}_v^{\text{inter}}$ contains nodes simultaneously close to the anchor in the embedding space and in the WL-structural space (marked **green** 🟢 in Figure 2). Then, we combine the intersection with the original positive set to form an extended positive set for anchor as:

$$\mathcal{P}_v^{\text{extend}} = \mathcal{P}_v \cup \mathcal{P}_v^{\text{inter}} \quad \forall v \in \mathcal{V}. \tag{7}$$

This converts the single-positive regime into a multi-positive regime, while keeping all positives cross-view to avoid trivial same-view matching.

Nodes that satisfy only one of the two criteria for similarity based on features or structure are designated as hard negatives:

$$\mathcal{N}_v^{\text{hard}} = (\mathcal{P}_v^{\text{feat}} \cup \mathcal{P}_v^{\text{struct}}) \setminus \mathcal{P}_v^{\text{inter}} \quad \forall v \in \mathcal{V}. \tag{8}$$

These nodes (marked **gray** ⬤ in Figure 2) are semantically harder than random negatives and naturally reflect misleading similarities, which makes them especially valuable for contrastive learning.

Finally, the extended positive set $\mathcal{P}_v^{\text{extend}}$ and the hard negative set $\mathcal{N}_v^{\text{hard}}$ are fed into our contrastive loss (described in the next section), where the extended sample space strengthens both the alignment and the uniformity properties of the learned representation.

### 3.5 Contrastive Objective with WL-Contrast

A crucial challenge in contrastive learning is the construction of informative negative samples. In InfoNCE-style (van den Oord et al., 2018) objectives, easy negatives contribute little to the gradient signal, whereas overly difficult ones may lead to training collapse. Prior work has explored enriching the negative set with harder samples, for instance, by synthesizing interpolated representations in the embedding space as in MoCHi (Kalantidis et al., 2020). In particular, given an anchor node $v$, a subset of hard negatives is first identified, and additional harder and hardest negatives are generated by linearly interpolating pairs of embeddings (e.g., hard-hard and anchor-hard pairs), yielding progressively more challenging samples.

In this work, we introduce WL-Contrast, a structure-aware contrastive objective tailored to dual-view graph representations. Our formulation departs from existing approaches in two key aspects. First, we move beyond the single-positive assumption by defining a multi-positive set $\mathcal{P}_v^{\text{extend}}$ (Eq. (7)), capturing nodes that are jointly consistent across both feature and structural views. Second, we propose a semantically grounded notion of hard negatives $\mathcal{N}_v^{\text{hard}}$ (Eq. (8)), defined as nodes that are similar in one view (feature or structure) but dissimilar in the other. Inspired by MoCHi, we construct progressively more challenging negatives via interpolation. Specifically, we generate additional sets of harder negatives, denoted as $\mathcal{N}_v^{\text{harder}}[k]$ and $\mathcal{N}_v^{\text{hardest}}[k']$, where $k$ and $k'$ indicate the selected number of harder and hardest negatives, respectively. These sets extend the initial hard negative pool and further strengthen the contrastive signal. Our proposed version of the WL-Contrast loss function becomes:

$$\mathcal{L}_{\text{WL-Contrast}}(v) = \frac{1}{|\mathcal{P}_v^{\text{extend}}|} \sum_{p \in \mathcal{P}_v^{\text{extend}}} -\log \frac{\exp(\cos(\mathbf{h}_v, \mathbf{h}_p)/\tau)}{\exp(\cos(\mathbf{h}_v, \mathbf{h}_p)/\tau) + \sum_{n \in \mathcal{N}_v^{\text{hard}}} \exp(\cos(\mathbf{h}_v, \mathbf{h}_n)/\tau) + \mathcal{L}_{\text{MoCHi}}(v)},$$
(9)

where $\exp(\cdot)$ is the exponential function, $\tau$ is a temperature parameter, and $\mathcal{L}_{\text{MoCHi}}(v)$ is calculated as:

$$\mathcal{L}_{\text{MoCHi}}(v) = \sum_{n \in \mathcal{N}_v^{\text{harder}}[k]} \exp(\cos(\mathbf{h}_v, \mathbf{h}_n)/\tau) + \sum_{n \in \mathcal{N}_v^{\text{hardest}}[k']} \exp(\cos(\mathbf{h}_v, \mathbf{h}_n)/\tau).$$
(10)

By defining hard negatives via cross-view inconsistency, our approach yields more informative and structure-aware negative samples, reducing sampling bias and improving contrastive discrimination. Together with the multi-positive formulation, this leads to a stronger and more stable self-supervised representation.

For a technical overview of our proposed WLGCL method, we report all steps involved in the pipeline in Algorithm 1.

### 3.6 Theoretical Insight and Complexity Analysis

From a geometric viewpoint, our dual-view sampling mechanism improves the fundamental trade-off between alignment and uniformity in contrastive learning (Wang & Isola, 2020). Conventional GCL models often emphasize alignment by pulling together nodes that are close in the learned embedding space, but these pairs may not be structurally consistent, leading to over-clustered embeddings and a loss of generalization. In contrast, our intersection-based positive selection enforces cross-view consistency between feature and structure, ensuring that only semantically reliable pairs contribute to alignment. At the same time, nodes that are similar in one view but dissimilar in the other are treated as hard negatives, which enhances uniformity by preserving structural diversity in the representation space. This balanced mechanism mitigates over-smoothing and false-positive alignment, resulting in embeddings that are both discriminative and topology-aware.

---

**Algorithm 1** WLGCL overall pipeline.

---

1: **Input:** Graph data $\mathcal{G} = \{\mathcal{V}, \mathcal{E}\}$, node features $\mathbf{X} \in \mathbb{R}^{|V| \times F}$, GNN encoder, iterations number for WL tree $T$, feature-based threshold $\theta_{\text{feat}}$, structure-based threshold $\theta_{\text{struct}}$, top-k value $k$.
2: **Output:** Loss function value $\mathcal{L}_{\text{WL-Contrast}}(v)$.
3: # WL Tree Refinement Process
4: **for** $t \in \{1, 2, \dots T\}$ **do**
5:     Compute $C_v^{(t)}$ according to (1).
6: **end for**
7: # Dual-View Candidate Retrieval
8: Given $\theta_{\text{feat}}$, compute $\mathcal{P}_v^{\text{feat}}$ according to (2) and (3).
9: Given $\theta_{\text{struct}}$ and $C_v^{(T)}$, compute $\mathcal{P}_v^{\text{struct}}$ according to (4) and (5).
10: # Intersection-Based Positive Extension and Hard Negative Mining
11: Compute $\mathcal{P}_v^{\text{extend}}$ according to (6) and (7).
12: Compute $\mathcal{N}_v^{\text{hard}}$ according to (8).
13: Given $k$, compute $\mathcal{N}_v^{\text{harder}}$ by linear interpolating anchor and negative samples.
14: Given $k'$, compute $\mathcal{N}_v^{\text{hardest}}$ by linear interpolating two random negative samples.
15: # Loss Function Computation
16: Calculate $\mathcal{L}_{\text{WL-Contrast}}(v)$ according to (9) and (10).
17: **Return:** $\mathcal{L}_{\text{WL-Contrast}}(v)$.

---

In terms of computational properties, the proposed framework introduces minimal additional cost and remains fully compatible with standard GCL training. The structure-based similarity derived from the WL refinement is precomputed only once before training and reused throughout all epochs, scales linearly with the number of nodes $\mathcal{V}$ and edges $\mathcal{E}$, which is $\mathcal{O}(\|\mathcal{V}\| + \|\mathcal{E}\|)$. The feature-based similarity, on the other hand, must be updated dynamically at each epoch, since it depends on the evolving encoder representations. Given node embeddings $\mathbf{Z} \in \mathbb{R}^{|\mathcal{V}| \times d}$, computing pairwise cosine similarities requires $\mathcal{O}(\|\mathcal{V}\|^2 d)$ operations in the worst case. The subsequent threshold-based candidate selection introduces negligible additional overhead. Therefore, the overall per-epoch complexity is dominated by the encoder computation and pairwise similarity evaluation. In practice, these operations are efficiently implemented using matrix multiplication and GPU parallelization, making the additional sampling overhead relatively small compared with the backbone training cost. Regarding memory consumption, the proposed method stores a precomputed WL structural similarity matrix and a feature similarity matrix during candidate retrieval, requiring $\mathcal{O}(\|\mathcal{V}\|^2)$ memory in the dense implementation. For very large-scale graphs, the feature-similarity retrieval stage can be further accelerated using approximate nearest-neighbor search techniques (e.g., locality-sensitive hashing or graph-based ANN indexing), which substantially reduce both computational and memory costs while preserving high-quality candidate selection. Although such approximations are not adopted in our current implementation, they are fully compatible with the proposed framework and provide a practical avenue for further scalability improvements. Importantly, our sampling module is model-agnostic and can be seamlessly integrated into any GCL backbone. Overall, the method achieves a favorable balance between principal structural design, semantic reliability, and practical scalability, providing an efficient and general plug-in for structure-aware GCL.

## 4 Experiments

### 4.1 Experimental Settings

**Baselines.** We conduct a comprehensive set of experiments to evaluate our proposed model against seven representative GCL baselines: DGI (Velickovic et al., 2019), GRACE (Zhu et al., 2020), BGRL (Thakoor et al., 2022), MUSE (Yuan et al., 2023), EPAGCL (Xu et al., 2025), FOSSIL (Sangare et al., 2025), and PolyGCL (Chen et al., 2024). These methods cover a broad spectrum of contrastive objectives, augmentation strategies, and structure-aware designs, providing a diverse and competitive comparison. Table 1 gives a

Table 1: High-level comparison of representative graph contrastive learning baselines.

| Model | Contrast Type | View Strategy | Negatives | Contrast Scale |
|---|---|---|---|---|
| DGI (Velickovic et al., 2019) | MI-based | Global vs local | ✓ | Node / Graph |
| GRACE (Zhu et al., 2020) | InfoNCE-based | Perturbation-based | ✓ | Node |
| BGRL (Thakoor et al., 2022) | Bootstrap | Perturbation-based | ✗ | Node |
| MUSE (Yuan et al., 2023) | Hybrid (contrast + recon.) | Feature-based | ✓ | Node |
| PolyGCL (Chen et al., 2024) | Spectral contrast | Frequency-domain | ✓ | Node |
| EPAGCL (Xu et al., 2025) | Contrastive regularization | Topology-based | ✓ | Node |
| FOSSIL (Sangare et al., 2025) | Structural alignment | Subgraph matching | ✗ | Subgraph |
| **WLGCL** | Structure-aware | Dual-view (feature + WL) | ✓ | Node |

Table 2: Statistics of the datasets used.

| | **Cora** | **CiteSeer** | **Amazon-Photo** | **Actor** | **Squirrel** | **Chameleon** |
|---|---|---|---|---|---|---|
| Nodes | 2,708 | 3,327 | 7,650 | 7,600 | 5,201 | 2,277 |
| Edges | 10,556 | 9,104 | 238,162 | 30,019 | 217,073 | 36,101 |
| Features | 1,433 | 3,703 | 745 | 932 | 2,089 | 2,325 |
| Classes | 7 | 6 | 8 | 5 | 5 | 5 |

brief summary and comparison between the baseline models and our proposed method. For all baselines, we use the default hyperparameters provided in their code.

**Datasets.** Experiments are performed on three homophilic datasets, Cora (Yang et al., 2016), Cite-Seer (Yang et al., 2016), and Amazon-Photo (Shchur et al., 2018), as well as three heterophilic datasets: Actor (Pei et al., 2020), Squirrel (Rozemberczki et al., 2021), and Chameleon (Rozemberczki et al., 2021). These together enable a thorough assessment of performance under varying levels of structural alignment. The statistics of the datasets we used are demonstrated in Table 2.

**Training and test paradigms.** For all our experiments, WLGCL uses WLHN (Nikolentzos et al., 2023) as the GNN backbone encoder architecture, using 4 layers and a hidden dimension of 64 as suggested in their paper. To isolate the effect of our proposed dual-view sampling and WL-Contrast objective, we also include a control baseline, where we follow the standard GRACE training pipeline but replace its encoder with WLHN (denoted as GCL+WLHN). The feature-similarity and structure-similarity thresholds $\theta_{\text{feat}}$ and $\theta_{\text{struct}}$, respectively, are tuned individually for each dataset to account for differences in graph density, feature quality, and homophily level. All models are trained and evaluated under identical conditions. Performance is assessed via downstream node classification accuracy, using a logistic regression classifier trained on the learned node embeddings. Importantly, the train/test masks are used only for this downstream evaluation and play no role in training the embedding model itself. The representation learning phase is entirely self-supervised: the GNN is trained solely to learn node embeddings, without using any label information or predefined data splits. When datasets provide predefined masks, these are adopted as-is for the evaluation stage.

For datasets without such masks, we randomly sample 20% of the nodes as the training set and use the remaining 80% for testing, with identical splits across all methods and random seeds. Every experiment was repeated five times with different random seeds and we report the mean value and standard deviation for the performance. Our experiments are conducted on a machine equipped with an NVIDIA RTX A5000 GPU, ensuring consistency and reproducibility across all baselines and datasets.

## 4.2 Results and Discussion

Table 3 reports the classification accuracy for node classification across six benchmark datasets. Our proposed WLGCL consistently matches or surpasses the strongest baseline on every dataset, demonstrating the effectiveness of jointly enforcing feature-structure consistency in contrastive learning. Furthermore, WLGCL steadily surpasses the GCL+WLHN baseline, highlighting that, despite sharing the same encoder architec-

Table 3: Node classification accuracy ($\pm$ std) comparison across six benchmark datasets over 5 runs. The best results are highlighted in **bold**, while the second best-performing methods are underlined. The improvement is calculated base on WLGCL and the second best-performing method.

| Model | Homophilic | | | Heterophilic | | |
|---|---|---|---|---|---|---|
| | Cora | CiteSeer | Amazon-Photo | Actor | Squirrel | Chameleon |
| GRACE | $0.671_{\pm0.025}$ | $0.558_{\pm0.009}$ | $0.709_{\pm0.048}$ | $0.258_{\pm0.014}$ | $0.309_{\pm0.013}$ | $0.383_{\pm0.026}$ |
| DGI | $0.704_{\pm0.055}$ | $0.535_{\pm0.033}$ | $0.628_{\pm0.044}$ | $0.256_{\pm0.015}$ | $0.308_{\pm0.032}$ | $0.397_{\pm0.067}$ |
| BGRL | $0.706_{\pm0.010}$ | $0.598_{\pm0.015}$ | $0.729_{\pm0.032}$ | $0.240_{\pm0.050}$ | $0.306_{\pm0.033}$ | $0.401_{\pm0.071}$ |
| MUSE | $0.673_{\pm0.033}$ | $0.603_{\pm0.023}$ | $0.736_{\pm0.028}$ | $0.279_{\pm0.034}$ | $0.294_{\pm0.045}$ | $0.400_{\pm0.103}$ |
| EPAGCL | $0.750_{\pm0.020}$ | $0.644_{\pm0.022}$ | $0.714_{\pm0.039}$ | $0.272_{\pm0.007}$ | $0.323_{\pm0.017}$ | $0.320_{\pm0.049}$ |
| FOSSIL | $\underline{0.758}_{\pm0.006}$ | $0.634_{\pm0.010}$ | $\underline{0.761}_{\pm0.024}$ | $\underline{0.343}_{\pm0.003}$ | $\underline{0.344}_{\pm0.014}$ | $\underline{0.432}_{\pm0.029}$ |
| PolyGCL | $0.751_{\pm0.011}$ | $\underline{0.665}_{\pm0.018}$ | $0.758_{\pm0.031}$ | $0.321_{\pm0.009}$ | $0.336_{\pm0.010}$ | $0.399_{\pm0.020}$ |
| GCL+WLHN | $0.736_{\pm0.006}$ | $0.629_{\pm0.019}$ | $0.718_{\pm0.037}$ | $0.323_{\pm0.034}$ | $0.317_{\pm0.029}$ | $0.393_{\pm0.069}$ |
| **WLGCL** | $\mathbf{0.764}_{\pm0.013}$ | $\mathbf{0.678}_{\pm0.010}$ | $\mathbf{0.774}_{\pm0.034}$ | $\mathbf{0.358}_{\pm0.014}$ | $\mathbf{0.357}_{\pm0.024}$ | $\mathbf{0.452}_{\pm0.036}$ |
| Improvement | +0.79% | +1.95% | +1.71% | +4.37% | +3.78% | +4.63% |
| $p$-value[†] | 0.039 | 0.056 | 0.047 | 0.020 | 0.028 | 0.019 |

*GCL+WLHN here denotes using WLHN as encoder within a standard GCL pipeline.*

[†]*Statistical significance is computed between best and second-best models through paired t-test (p-value < 0.05).*

ture, our method incorporates additional design choices that substantially enhance predictive performance and contribute to its superior accuracy. On the homophilic graphs Cora, CiteSeer, and Amazon-Photo, WLGCL yields clear improvements over the best-performing baselines. On Cora, WLGCL achieves an accuracy of 0.764, outperforming FOSSIL (0.758) by +0.79%. On CiteSeer, WLGCL reaches 0.678, surpassing PolyGCL (0.665) by +1.95%. On Amazon-Photo, the improvement is even more pronounced, outperforming FOSSIL (0.761) by +1.71%. These gains show that even in feature-homophilic settings, incorporating structural consistency leads to better positive/negative sample construction and more discriminative embeddings.

The advantages become even clearer on heterophilic datasets, where feature-based similarity alone is unreliable for sampling due to the mismatch between node features and structural roles. Across three heterogeneous datasets, our model achieved an improvement of approximately 4%. These substantial improvements can be attributed to the fact that, in heterophilic graphs, structurally similar nodes often exhibit dissimilar features, causing feature-only methods to produce misleading positives and weak negatives. By explicitly incorporating WL-based structural similarity, our framework is able to recover such structurally consistent relationships, while the dual-view sampling strategy filters out inconsistent pairs and identifies informative hard negatives. This leads to more reliable contrastive signals and ultimately stronger representation learning when topology carries more semantic information than raw features.

To further assess whether these improvements are statistically meaningful, we conducted paired statistical significance tests between WLGCL and the best baseline method using the results obtained from five random seeds. As reported in the $p$-value row of Table 3, the improvements are statistically significant ($p < 0.05$) on five out of six datasets. On CiteSeer, although the $p$-value slightly exceeds the conventional significance threshold ($p = 0.056$ vs. 0.05), the margin is very limited, and WLGCL still achieves the highest average accuracy among all compared methods. These results demonstrate that the superiority of WLGCL is not due to random fluctuations introduced by different initialization seeds.

Overall, the results demonstrate that our structure-informed, multi-positive sampling framework provides consistent and often significant performance gains over both classical and state-of-the-art GCL baselines.

### 4.3 Ablation Study

**Impact of encoder choice.** To analyze the importance of the encoder architecture in our framework, we compare the performance of four different GNN backbones: GCN (Kipf & Welling, 2017), GIN (Xu et al., 2019), GAT (Velickovic et al., 2018), and WLHN (Nikolentzos et al., 2023), which is our chosen backbone. All baseline encoders are configured with two layers, hidden dimension 64, and output dimension 32 to

Table 4: Node classification accuracy ($\pm$ std) of different encoders across six benchmark datasets over 5 runs.

| Encoder | Homophilic | | | Heterophilic | | |
|---|---|---|---|---|---|---|
| | Cora | CiteSeer | Amazon-Photo | Actor | Squirrel | Chameleon |
| GCN | $\underline{0.735}_{\pm 0.010}$ | $0.568_{\pm 0.025}$ | $0.770_{\pm 0.023}$ | $0.306_{\pm 0.014}$ | $\underline{0.355}_{\pm 0.024}$ | $0.415_{\pm 0.026}$ |
| GIN | $0.711_{\pm 0.021}$ | $\underline{0.577}_{\pm 0.029}$ | $0.755_{\pm 0.032}$ | $0.290_{\pm 0.018}$ | $0.337_{\pm 0.033}$ | $0.398_{\pm 0.088}$ |
| GAT | $0.725_{\pm 0.010}$ | $0.559_{\pm 0.028}$ | $\underline{0.772}_{\pm 0.032}$ | $\underline{0.321}_{\pm 0.011}$ | $0.325_{\pm 0.054}$ | $\underline{0.423}_{\pm 0.057}$ |
| WLHN | $\mathbf{0.764}_{\pm 0.013}$ | $\mathbf{0.678}_{\pm 0.010}$ | $\mathbf{0.774}_{\pm 0.034}$ | $\mathbf{0.358}_{\pm 0.014}$ | $\mathbf{0.357}_{\pm 0.024}$ | $\mathbf{0.435}_{\pm 0.036}$ |

ensure a fair comparison, while WLHN follows the parameters described in Section 4.1. These encoders are integrated into our dual-view contrastive learning pipeline without modifying any other component.

Table 4 reports the node classification accuracy across six datasets. The results show that WLHN consistently outperforms the other encoders on all datasets, demonstrating its advantage in capturing structural nuances that are crucial to our feature-structure intersection mechanism. For example, on heterophilic datasets such as Actor, Squirrel, and Chameleon, WLHN achieves accuracy improvements of +11.9%, +9.8%, and +5.3% over the strongest alternative encoder, respectively. Even on homophilic datasets (Cora, CiteSeer, Amazon-Photo), WLHN provides noticeable gains, achieving the highest accuracy in all cases.

These results validate that the choice of encoder plays a critical role in the effectiveness of WLGCL. Because WLHN generates richer structural representations and preserves multi-scale neighborhood information, it aligns naturally with our structure-aware positive and hard negative sampling strategy, leading to superior contrastive representations. Importantly, we also observe that the proposed framework consistently achieves strong performance relative to the baselines on heterophilic datasets across different encoder choices. This suggests that the observed improvements are primarily attributed to the effectiveness of the dual-view contrastive framework itself, rather than being dependent on a specific encoder architecture.

**Components of the contrastive objective.** To better understand the contribution of each component in the proposed WL-Contrast objective, we perform a component-wise ablation study by independently analyzing the positive and negative sampling mechanisms while keeping all other training settings unchanged. In all experiments, the WLHN encoder is fixed to eliminate the influence of backbone architecture.

For the positive sampling study, we ignore the hard-negative construction and compare four positive-selection strategies: (1) *No Expanding*, which only uses the augmentation counterpart of the anchor node as the positive sample, equivalent to the standard contrastive setting; (2) *Feature-only*, where additional positives are selected solely according to feature similarity; (3) *Structure-only*, where positives are selected solely according to WL-based structural similarity; and (4) *Intersection*, where only nodes that are simultaneously similar in both feature and structural views are treated as additional positives.

As shown in Table 5, simply expanding positives using feature similarity provides only marginal improvements over the standard setting and can even slightly degrade performance on heterophilic datasets. In contrast, WL-based structural positives consistently outperform feature-only positives, indicating that structural consistency provides complementary information beyond latent feature similarity. The best results are achieved by the intersection strategy across all datasets. This observation suggests that requiring agreement between the two views effectively filters noisy candidates and produces a higher-quality positive set, leading to more reliable representation alignment.

For the hard-negative sampling study, we fix the positive expansion strategy and compare three hard-negative constructions: (1) *Feature-only*, where hard negatives are mined exclusively from feature-similar candidates; (2) *Structure-only*, where hard negatives are selected from structurally similar candidates identified by the WL view; and (3) *Dual-branch*, which combines hard negatives from both views. The results show that both feature-based and structure-based hard negatives improve performance, confirming that challenging negatives contribute useful contrastive signals. More importantly, combining the two sources consistently

Table 5: Node classification accuracy ($\pm$ std) of component-wise contrastive loss function on four datasets over 5 runs.

| Sampling | Strategies | Cora | CiteSeer | Actor | Squirrel |
|---|---|---|---|---|---|
| Positive | No Expanding | $0.703_{\pm 0.020}$ | $0.616_{\pm 0.028}$ | $0.318_{\pm 0.043}$ | $0.308_{\pm 0.025}$ |
| | Feature-only | $0.713_{\pm 0.018}$ | $0.624_{\pm 0.011}$ | $0.298_{\pm 0.035}$ | $0.299_{\pm 0.055}$ |
| | Structure-only | $0.704_{\pm 0.015}$ | $0.619_{\pm 0.023}$ | $0.321_{\pm 0.058}$ | $0.332_{\pm 0.053}$ |
| | Intersection | $\mathbf{0.734}_{\pm 0.010}$ | $\mathbf{0.633}_{\pm 0.015}$ | $\mathbf{0.339}_{\pm 0.042}$ | $\mathbf{0.342}_{\pm 0.067}$ |
| Negative | Feature-only | $0.746_{\pm 0.010}$ | $0.651_{\pm 0.016}$ | $0.334_{\pm 0.049}$ | $0.335_{\pm 0.060}$ |
| | Structure-only | $0.742_{\pm 0.037}$ | $0.643_{\pm 0.022}$ | $0.341_{\pm 0.040}$ | $0.344_{\pm 0.050}$ |
| | Dual-branch | $\mathbf{0.764}_{\pm 0.013}$ | $\mathbf{0.678}_{\pm 0.010}$ | $\mathbf{0.358}_{\pm 0.014}$ | $\mathbf{0.357}_{\pm 0.024}$ |

yields the best performance on all datasets. This demonstrates that the two views capture complementary forms of node similarity and that exploiting their disagreement provides a richer and more informative hard-negative set than either view alone.

Overall, the results further support that both components of WL-Contrast are essential. The intersection-based positive expansion improves the reliability of positive samples, while dual-view hard-negative mining provides stronger contrastive supervision. Their combination leads to the most consistent gains across both homophilic and heterophilic graphs.

**Sampling quality of different strategies.** To better understand the semantic properties of the proposed sampling strategy, we measure the label agreement between each sampled node and its anchor node. Specifically, label agreement is defined as the proportion of sampled node pairs sharing the same ground-truth label. Although labels are not used during training, this analysis provides a useful post-hoc indicator of the semantic consistency of different candidate sets.

Results in Table 6 show that the intersection positives $\mathcal{P}^{\mathrm{inter}}$ consistently achieve the highest label agreement across all datasets. These results indicate that requiring consistency in both feature and structural views effectively filters out noisy candidates and yields a substantially more reliable positive set. The comparison between $\mathcal{P}^{\mathrm{feat}}$ and $\mathcal{P}^{\mathrm{struct}}$ further reveals that the two views capture complementary information. Feature-based retrieval generally exhibits higher semantic consistency on homophilic datasets, while structural retrieval often identifies nodes with similar structural roles rather than identical class semantics. Interestingly, on the heterophilic datasets, structural positives mainly achieve substantially higher agreement than feature positives (e.g. 90.9% vs. 41.9% on Squirrel), suggesting that WL-based structural similarity captures discriminative information that is not available from node features alone. So that the intersection operation acts as a precision-enhancing mechanism that preserves candidates supported by both views.

We further analyze the proposed hard negatives. The feature-only hard negatives $\mathcal{N}^{\mathrm{feat}}$ maintain relatively high label agreement on homophilic datasets (e.g., 66.7% on Cora), indicating that they remain semantically related to the anchor node and therefore constitute informative hard negatives rather than trivial negatives. In contrast, $\mathcal{N}^{\mathrm{struct}}$ consists of nodes that are considered structurally similar by the WL view but are not supported by feature similarity. The relatively low agreement observed on Cora (23.9%) and CiteSeer (17.4%) suggests that these nodes often share structural patterns or functional roles with the anchor while belonging to different semantic classes. This observation highlights the complementary nature of the structural view: rather than merely recovering label-consistent neighbors, it captures additional structural information that is not reflected in node attributes. Consequently, treating these cross-view inconsistent samples as hard negatives encourages the model to distinguish structural resemblance from semantic consistency, leading to more discriminative representations. The combined hard-negative set $\mathcal{N}^{\mathrm{dual}}$ consistently exhibits intermediate agreement levels between positives and random negatives, indicating that it provides a challenging yet informative contrastive signal.

Finally, on heterophilic datasets such as Squirrel and Chameleon, the structural view already achieves high agreement, while the intersection criterion further improves the purity of the positive set. This observation

Table 6: Label agreement of different sampling strategy on six datasets over 5 runs. $\mathcal{P}$ and $\mathcal{N}$ stand for expanding positive and hard negative sets.

| | Homophilic | | | Heterophilic | | |
|---|---|---|---|---|---|---|
| | Cora | CiteSeer | Amazon-Photo | Actor | Squirrel | Chameleon |
| $\mathcal{P}^{\text{feat}}$ | $0.820_{\pm0.020}$ | $0.530_{\pm0.013}$ | $0.858_{\pm0.060}$ | $0.257_{\pm0.003}$ | $0.419_{\pm0.057}$ | $0.665_{\pm0.019}$ |
| $\mathcal{P}^{\text{struct}}$ | $0.461_{\pm0.000}$ | $0.204_{\pm0.000}$ | $0.506_{\pm0.000}$ | $0.278_{\pm0.000}$ | $0.909_{\pm0.000}$ | $0.934_{\pm0.000}$ |
| $\mathcal{P}^{\text{inter}}$ | $0.917_{\pm0.024}$ | $0.605_{\pm0.007}$ | $0.776_{\pm0.165}$ | $0.305_{\pm0.019}$ | $0.988_{\pm0.005}$ | $0.981_{\pm0.007}$ |
| $\mathcal{N}^{\text{feat}}$ | $0.667_{\pm0.092}$ | $0.339_{\pm0.038}$ | $0.550_{\pm0.071}$ | $0.240_{\pm0.007}$ | $0.261_{\pm0.052}$ | $0.428_{\pm0.108}$ |
| $\mathcal{N}^{\text{struct}}$ | $0.239_{\pm0.006}$ | $0.174_{\pm0.002}$ | $0.237_{\pm0.005}$ | $0.210_{\pm0.002}$ | $0.858_{\pm0.015}$ | $0.812_{\pm0.010}$ |
| $\mathcal{N}^{\text{dual}}$ | $0.464_{\pm0.047}$ | $0.256_{\pm0.019}$ | $0.429_{\pm0.043}$ | $0.226_{\pm0.003}$ | $0.493_{\pm0.035}$ | $0.607_{\pm0.058}$ |

highlights the importance of incorporating structural information into contrastive sampling and helps explain the superior performance of WLGCL on heterophilic graphs.

## 4.4 Time Efficiency Trade-off

To further analyze the practicality of our method, we evaluate its time efficiency relative to seven representative GCL baselines. Figure 3 reports the trade-off between accuracy and training time per epoch (in seconds) on four datasets. Models such as GRACE, DGI, and BGRL achieve fast training speed due to their lightweight objectives, but their accuracy is considerably lower, especially on heterophilic graphs. In contrast, methods that incorporate more complex augmentations or contrastive processing, such as EPAGCL, PolyGCL, FOSSIL, obtain higher accuracy but require substantially longer training time.

Across all datasets, WLGCL achieves a favorable balance between computational cost and predictive performance. Although the dual-view sampling introduces moderate overhead, primarily from computing feature similarity each epoch, our framework remains competitive in runtime while consistently delivering the highest accuracy. Notably, on Cora and CiteSeer, WLGCL lies close to the Pareto frontier: it outperforms EPAGCL and PolyGCL by a large margin in accuracy while taking less time per epoch. The efficiency gain is even more evident on heterophilic datasets such as Actor and Squirrel, where our model achieves the best accuracy with only modest additional cost compared to the strongest baselines.

It is also important to note that the structure-based similarity branch, implemented through WL tree construction, is computed only once during a preprocessing stage. This cost is therefore not included in the per-epoch measurements shown in Figure 3. In practice, this preprocessing step is lightweight: its runtime is roughly equivalent to a single training epoch (0.15s on Cora, 0.21s on CiteSeer, 0.20s on Actor, and 0.15s on Squirrel). Because this cost is paid only once at the beginning of training, its amortized impact on overall efficiency is negligible. Overall, these results suggest that the proposed WLGCL provides significant accuracy improvements without incurring prohibitive computational overhead, making it suitable for real-world graph learning scenarios where both efficiency and performance are crucial.

## 4.5 Hyper-parameter Sensitivity

**Impact of feature similarity space.** We also examine the impact of different strategies for computing feature similarity in the dual-view sampling process. Specifically, we compare two variants: (i) Graph-space similarity, where node similarity is computed directly from the raw input features; (ii) Embedding-space similarity, where similarity is measured in the latent space produced by the WLHN encoder. Both variants share the same structure-based similarity branch and hard-negative construction, isolating the effect of the feature-similarity module. As shown in Table 7, using embedding-space similarity consistently leads to higher performance across all datasets. This behavior is expected: raw features may be noisy, sparse, or weakly aligned with structural patterns, making similarity estimates unreliable. In contrast, similarities measured in the learned latent space capture more discriminative and structure-aware representations, resulting in more accurate positive selection and more meaningful hard negatives. Overall, these results confirm that

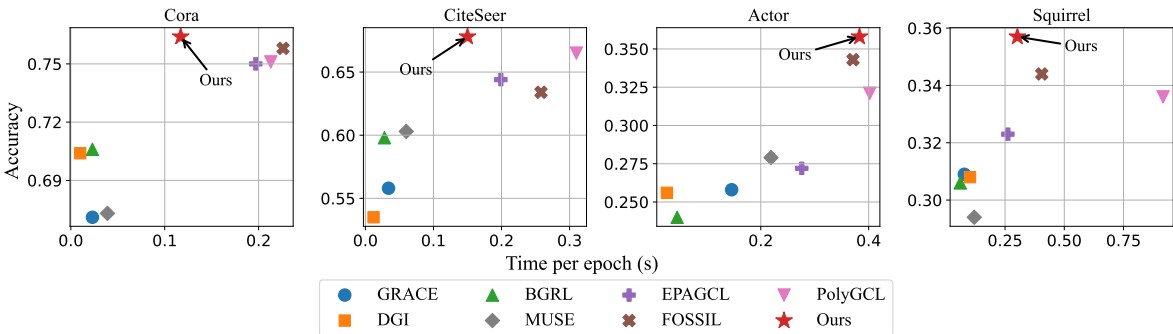

Figure 3: Time–accuracy comparison across four datasets. WLGCL ("Ours") attains higher accuracy with competitive computational cost.

Table 7: Node classification accuracy ($\pm$ std) of different feature similarity space over 5 runs.

| Feature Space | Cora | CiteSeer | Amazon-Photo | Actor | Squirrel | Chameleon |
|---|---|---|---|---|---|---|
| Graph Space | $0.759_{\pm 0.017}$ | $0.670_{\pm 0.021}$ | $0.760_{\pm 0.027}$ | $0.351_{\pm 0.014}$ | $0.349_{\pm 0.012}$ | $0439_{\pm 0.028}$ |
| Embedding Space | $\mathbf{0.764}_{\pm 0.013}$ | $\mathbf{0.678}_{\pm 0.010}$ | $\mathbf{0.774}_{\pm 0.034}$ | $\mathbf{0.358}_{\pm 0.014}$ | $\mathbf{0.357}_{\pm 0.024}$ | $\mathbf{0.452}_{\pm 0.036}$ |
| Improvement | $+0.66\%$ | $+1.19\%$ | $+2.24\%$ | $+1.99\%$ | $+2.29\%$ | $+2.96\%$ |

embedding-space similarity provides a stronger semantic signal, thereby enhancing the effectiveness of our dual-view contrastive framework.

**Effect of WL tree depth.** We finally investigate how the depth of the WL tree affects the structure-based similarity branch in our dual-view framework. Figure 4 reports the performance of our model when varying the WL refinement depth from two to six. We observe a consistent pattern across all datasets: accuracy improves as the depth increases from two to around four, after which performance begins to plateau or slightly decline. This trend aligns with the intuition behind WL refinement. Shallow depths capture only very local structural patterns, leading to coarse or noisy structural similarities. As a result, the structure-based similarity becomes less informative, leading to suboptimal intersections with the feature-based positives. Increasing the depth expands the receptive field and allows the WL subtree to encode richer multi-hop dependencies, enabling more meaningful structural matches and better alignment with the embedding space. However, excessively increasing the WL depth introduces a different limitation. As refinement proceeds, WL signatures become increasingly fine-grained and discriminative: nodes that differ at any smaller depth will never become identical at larger depths, and their structural distance can only increase or remain as zero. As a result, deep WL refinement fragments the node space into many highly specific structural equivalence classes, substantially reducing the number of structurally similar node pairs. This sparsification of structural positives limits their intersection with feature-based positives and weakens the contrastive learning signal between the two views. Overall, a WL depth of approximately four provides the best balance between local and global structural information and yields the strongest performance within our dual-view contrastive learning framework.

## 5 Conclusion and Future Work

In this work, we introduced WLGCL, a dual-view graph contrastive learning framework that explicitly integrates feature similarity and WL-based structural similarity. By disentangling similarity into two complementary perspectives, our method constructs a more discriminative set of positives through feature-structure intersection, and generates semantically meaningful hard negatives through cross-view inconsistency. Combined with the WL-Contrast objective, WLGCL achieves more robust alignment and better uniformity in the latent space. Extensive experiments on both homophilic and heterophilic datasets demonstrate that our

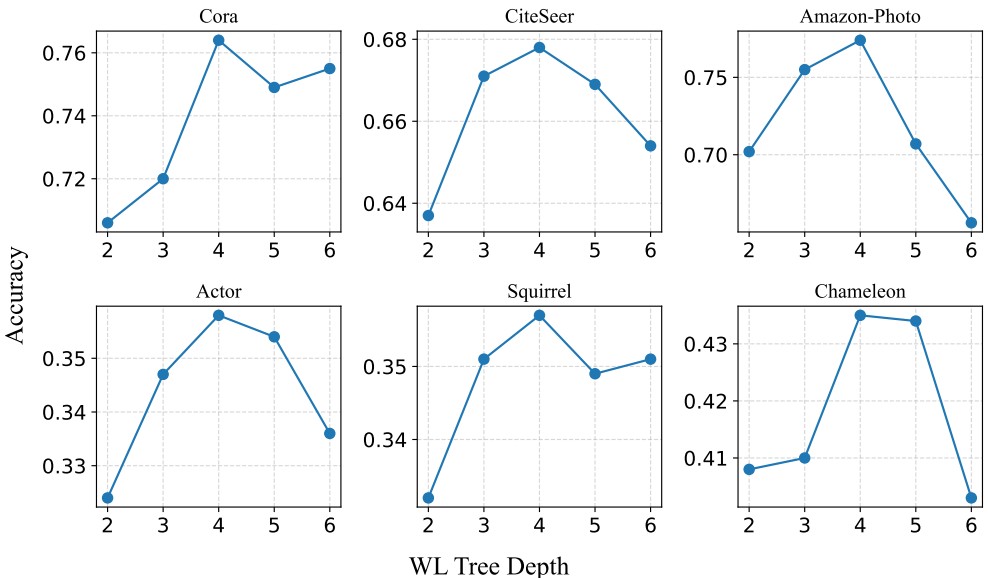

Figure 4: Node classification accuracy under different WL tree depths across six datasets.

method consistently outperforms state-of-the-art GCL models in accuracy while maintaining competitive training efficiency. Further ablation studies and hyper-parameter sensitivity analyses confirm the benefits of dual-view sampling, multi-positive contrastive objectives, embedding-space similarity, and WL tree depth optimization.

Looking forward, several promising directions remain for future work. First, while WLGCL relies on WL refinement for structural similarity, more expressive structural priors such as motif-based patterns or optimal transport distances could be explored to further enrich structure-aware contrastive sampling. Second, the proposed dual-view framework can be extended beyond node-level learning to subgraph- and graph-level representation learning, enabling applications in molecular graphs and other structured domains. Finally, improving the scalability of WL-based similarity computation and establishing stronger theoretical guarantees for structure-aware contrastive objectives remain important directions for future investigation.

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

# A  Hyperparameter Search Space and Tuning Procedure

To evaluate the sensitivity of the proposed method to the similarity thresholds, we perform dataset-specific hyperparameter tuning for the feature similarity threshold $\theta_{\text{feat}}$ and the structural similarity threshold $\theta_{\text{struct}}$. The hidden layer dimension and similarity thresholds are selected using a validation set, and the same tuning protocol is applied consistently across all datasets.

The search space of the hidden layer dimension is defined as $d_{\text{hidden}} \in \{32, 64, 128\}$. Then, for the structure similarity threshold, we search over $\theta_{\text{struct}} \in \{4, 6, 8, 10\}$. The range of $\theta_{\text{struct}}$ is determined based on the maximum Weisfeiler–Lehman (WL) tree depth used in our experiments. Since the maximum WL depth is 6, the maximum possible tree distance is 12. Therefore, the selected search range covers a sufficiently broad interval around the theoretically meaningful distance scale.

The feature similarity threshold $\theta_{\text{feat}}$ is dataset-dependent due to variations in the distribution of structural similarity scores. To avoid using an unnecessarily large or small search space, we first analyze the empirical distribution of feature similarity values on each dataset. Table 8 summarizes the minimum, mean, median, and maximum values for each dataset. Based on these statistics, we construct dataset-specific search grids for $\theta_{\text{feat}}$. Specifically, the candidate values are sampled with step sizes of either 0.005 or 0.01 depending on the scale and density of the similarity distribution $\theta_{\text{feat}} \in \{a, a + \Delta, a + 2\Delta, \ldots, b\}$, where $a$ and $b$ are determined according to the observed similarity range, and $\Delta \in \{0.005, 0.01\}$.

For each dataset, all combinations of $(d_{\text{hidden}}, \theta_{\text{feat}}, \theta_{\text{struct}})$ are evaluated on the validation set, and the configuration achieving the best validation performance is selected for final test evaluation.

Table 8: Statistics of feature similarity value on six different datasets.

|        | Cora    | CiteSeer | Amazon-Photo | Actor   | Squirrel | Chameleon |
|--------|---------|----------|--------------|---------|----------|-----------|
| min    | -0.0185 | -0.0183  | -0.0126      | -0.0054 | -0.0090  | -0.0150   |
| mean   | 0.0054  | 0.0086   | 0.0311       | 0.0354  | 0.0329   | 0.0089    |
| median | 0.0040  | 0.0071   | 0.0312       | 0.0351  | 0.0330   | 0.0076    |
| max    | 0.0591  | 0.0638   | 0.0642       | 0.0643  | 0.0634   | 0.0609    |

# B  Statistics of Positive and Negative Candidate Sets

To further analyze the behavior of the proposed contrastive learning framework, we report the statistics of the positive and negative candidate sets generated during the validation stage. Specifically, we measure the total number of feature-based positive pairs $|\mathcal{P}_v^{\text{feat}}|$, structural-based positive pairs $|\mathcal{P}_v^{\text{struct}}|$, intersection positive pairs $|\mathcal{P}_v^{\text{inter}}|$, and hard negative pairs $|\mathcal{N}_v^{\text{hard}}|$. These statistics provide a quantitative analysis of the contrastive sampling behavior and the strength of the resulting training signals. The feature-based and structural-based positive sets capture complementary similarity relationships, while their intersection contains high-confidence positive pairs satisfying both criteria. The hard negative set contains challenging samples that are similar under one aspect but excluded from the positive set, providing informative negative signals for contrastive learning. Table 9 summarizes the candidate set statistics on different datasets. The reported values are collected during the validation stage using the selected hyperparameter configuration for each dataset.

Table 9: Statistics of positive and negative candidate sets during validation. The values denote the total number of sampled node pairs.

|                               | Cora   | CiteSeer | Amazon-Photo | Actor   | Squirrel | Chameleon |
|-------------------------------|--------|----------|--------------|---------|----------|-----------|
| $|\mathcal{P}_v^{\text{feat}}|$   | 30,500 | 198,580  | 39,048       | 69,158  | 7,998    | 3,266     |
| $|\mathcal{P}_v^{\text{struct}}|$ | 16,428 | 273,407  | 21,778       | 89,486  | 13,373   | 9,003     |
| $|\mathcal{P}_v^{\text{inter}}|$  | 556    | 10,304   | 1,938        | 2,130   | 2,580    | 1,118     |
| $|\mathcal{N}_v^{\text{hard}}|$   | 45,816 | 451,379  | 564,950      | 154,384 | 16,211   | 10,533    |

