# OpenReview forum: "Graph Contrastive Learning via Weisfeiler-Leman Dual-View Sampling"
_TMLR — Under review for TMLR_

### Review · Reviewer_PLDT · 2026-05-08

**Summary Of Contributions:**

The paper proposes WLGCL, a graph contrastive learning method that makes the choice of positive and negative training pairs more structure-aware. The authors argue that relying only on feature or embedding similarity can be unreliable, especially when node features do not align well with graph structure, as in heterophilic graphs.
The method combines two views of similarity: feature-based similarity from learned node embeddings and structure-based similarity from Weisfeiler–Leman refinement. For each anchor node, nodes that are similar in both views are treated as reliable positives, while nodes similar in only one view are treated as hard negatives.
The paper also introduces a multi-positive contrastive objective, so each node can learn from several positive examples rather than a single pair. Experiments on six node-classification benchmarks show improved performance over several graph contrastive learning baselines, with larger gains on heterophilic datasets.

**Audience:**

Yes

**Audience Explanation:**

Yes. At least some readers in TMLR’s audience would likely be interested in this paper, especially those working on graph representation learning, graph contrastive learning, self-supervised learning, and heterophilic graphs. The paper studies a concrete problem in GCL: feature-based similarity can be unreliable when node features and graph structure disagree, and it proposes a simple structure-aware alternative using Weisfeiler–Leman refinement to select positives and hard negatives. This is a specialized contribution, but still of interest to some.

**Broader Impact Concerns:**

The work is methodological and evaluated on standard node-classification benchmarks, so the immediate ethical risk is limited. However, if the method is later applied to social, biological, financial, or recommendation graphs, improved representation learning could affect sensitive decisions involving people or groups. The authors could briefly mention that structure-aware graph learning may amplify existing biases in graph topology or node attributes, especially when labels or connections reflect social or institutional bias.

**Claims And Evidence:**

No

**Claims Explanation:**

The claims are partly supported. The strongest evidence is for the basic performance claim: WLGCL is tested on six node-classification datasets, compared with several baselines, repeated over five random seeds, and reported with mean and standard deviation. The results show WLGCL doing best on all six datasets, with larger gains on the heterophilic ones.
The paper also gives some support for the claim that the WL-based contrastive design helps. The ablation shows that the proposed WL-Contrast loss performs better than InfoNCE and MoCHi on the tested datasets.
However, the broader explanation is less fully supported. The authors say their method creates more reliable positives and better hard negatives, but they mostly show this indirectly through accuracy. They do not directly measure whether the sampled positives are actually better, whether false positives are reduced, or whether alignment and uniformity improve. So this mechanism is plausible, but not completely proven.
Overall, the paper supports a narrower claim: WLGCL performs better than the selected baselines on these six benchmarks. The stronger claims about why it works, how generally it helps heterophilic graphs, and how practical it is need more direct evidence.

**Requested Changes:**

The authors should make the claims more precise and less broad. The experiments support the claim that WLGCL performs well on the six tested node-classification datasets, but they do not fully prove stronger claims about reliable positives, better hard negatives, alignment, uniformity, or general real-world usefulness.
Critical for acceptance: The paper should directly test whether the proposed sampling strategy really produces better positive and negative pairs. The central idea is that nodes similar in both feature space and WL structural space should make more reliable positives, while nodes similar in only one of the two spaces should make useful hard negatives. At the moment, this is mostly supported indirectly through better classification accuracy. The authors should add direct pair-quality checks. For example, using labels only for evaluation, they could measure how often selected positives share the same class as the anchor, compare feature-only positives, WL-only positives, and intersection positives, and test whether the proposed hard negatives are closer or more informative than random negatives while still belonging to different classes. They could also report these diagnostics separately on homophilic and heterophilic datasets, since the method’s motivation is strongest when feature similarity and graph structure disagree.

Critical for acceptance: The ablation study should separate the main components more clearly. The current results show that WL-Contrast performs better than InfoNCE and MoCHi, but it is still hard to tell which part of the method is responsible for the gain. The authors should add ablations that remove or isolate each component. For example, they could compare: feature-only positives, WL-only positives, intersection positives, extended positives without hard negatives, hard negatives without extended positives, feature-disagreement hard negatives only, structure-disagreement hard negatives only, and the full model. They could also report whether each variant improves performance consistently across homophilic and heterophilic datasets. This would make it much clearer whether the improvement comes from WL-based positives, multi-positive training, hard-negative mining, or the combination of all components.

Critical for acceptance: The empirical comparison should be statistically stronger. A critical difference diagram would be useful because the paper compares many methods across several datasets, and it would show whether WLGCL is statistically distinguishable from the baselines rather than only having the best average numbers. However, since the paper uses only six datasets, the authors should also report paired per-dataset differences, confidence intervals, and win/loss/tie counts against the strongest baselines. This would make the performance claim more reliable and easier to interpret.

---

> ### Author Response · Authors · 2026-06-26
>
> We thank the reviewer for their thoughtful evaluation and the requested changes, which we have incorporated in the revised version of the paper. Below, we provide detailed responses to the reviewer’s comments.
>
> 1. We added a new pair-quality analysis in Section 4.3 (Table 6) using ground-truth labels for evaluation. Specifically, we compare feature-based, structure-based, and intersection-based positives, as well as different hard-negative construction strategies, on both homophilic and heterophilic datasets. The results show that the proposed intersection positives generally achieve the highest label agreement, indicating more reliable positive pairs, while the dual-view hard negatives provide a better balance between difficulty and semantic distinction than single-view alternatives. These findings provide direct empirical evidence supporting the effectiveness of our proposed sampling strategy and complement the downstream classification results.
>
> 2. We agree that the original ablation study did not fully evaluate the contributions of the individual components in our contrastive objective. We added a new component-wise ablation study in Section 4.3 (Table 5), separately analyzing the effects of positive expansion and hard-negative construction. Specifically, we compare feature-only, structure-only, and intersection-based positive sampling, as well as feature-only, structure-only, and dual-branch hard negatives. The results show that neither the feature view nor the structural view alone achieves the best performance. Instead, the intersection-based positives consistently outperform single-view positives, while the dual-branch hard negatives achieve the strongest results among all negative sampling variants. These trends hold across both homophilic and heterophilic datasets, providing direct evidence that the improvements arise from the combination of WL-based structural information and dual-view contrastive sampling, rather than from any individual component alone.
>
> 3. We agree that statistical significance analysis can provide a stronger assessment. We revised Section 4.2 and Table 3 to include paired $t$-tests between WLGCL and the strongest baseline in each dataset. The resulting $p$-values are 0.039, 0.056, 0.047, 0.020, 0.028, and 0.019 for Cora, CiteSeer, Amazon-Photo, Actor, Squirrel, and Chameleon, respectively. These results indicate that the improvements achieved by WLGCL are statistically significant on five out of the six datasets at the 0.05 significance level, with the improvement on CiteSeer remaining close to statistical significance. We believe these additional results provide stronger evidence that the observed performance gains are not solely due to random variation and further support our empirical findings.

---

### Review · Reviewer_ZbAd · 2026-05-10

**Summary Of Contributions:**

The paper proposes to enhance graph contrastive learning by incorporating a structural similarity measure based on the WL graph decomposition.

**Audience:**

Yes

**Audience Explanation:**

Graph contrastive learning is an active area of research with established benchmarks. This work contributes competetive or SOTA results on this, and is undoubtedly interesting for this community.

**Broader Impact Concerns:**

No issues

**Claims And Evidence:**

Yes

**Claims Explanation:**

The paper claims to separate the node sampling by feature and structure similarity using the WL method for improved performance. The results show a convincing set of benchmarks, where the method gives minor, but consistent improvements. The method is also ablated against the encoder and loss types, which show how the model choices contribute to the performance.

**Requested Changes:**

My main concern with the paper is the hand-wavy nature and motivation of using structural contrasts in the first place, and using the WL method specifically. WL was designed as a way to annotate the graph structure into subgraph patterns to facilitate graph isomorphism computation between two graphs. In this paper WL is used to analyse structural similarity of nodes within one graph, and I'm not sure how this relates to the actual WL formalism, or what is the motivation for this. What specific problem the WL addresses? This hasn't been formalised, and it seems that WL is used as just a black-box method that gives some, not very well understood, notion of node similarity which turns out to work well enough in practise. I found the motivation of 3.0 and introduction to be hand-wavy, and lack rigor.

In the figure 1 it seems that the WL simply contributes a notion of node similarity that finds all other nodes with 1 neighbor. Ok, but do we then need the entire WL stack to figure this out? Why do we want to limit our focus on the 1-neighbor nodes in this example?

---

> ### Author Response · Authors · 2026-06-26
>
> We thank the reviewer for their thoughtful evaluation and the requested changes, which we have incorporated in the revised version of the paper. Below, we provide detailed responses to the reviewer’s comments.
>
> 1. Regarding the necessity of structural contrast, we have revised Section 1 (page 2) to more explicitly discuss a limitation of existing graph contrastive learning methods. Most GCL approaches construct positive pairs primarily according to feature-space similarity. However, in many real-world graphs, especially heterophilic graphs, feature similarity does not necessarily correspond to structural or semantic consistency. Therefore, feature-based sampling may incorrectly align nodes with different structural roles, while it may ignore structurally similar nodes with dissimilar features. We now formalize this issue as the motivation for introducing an additional structural view during contrastive pair construction.
>
> 2. Regarding the use of WL specifically, we revised Section 2.2 (pages 3-4) to clarify the connection between the original WL formalism and our node-level structural similarity objective. Although WL was originally proposed for graph isomorphism testing, its iterative color-refinement process naturally induces rooted-subtree signatures for individual nodes. Through repeated neighborhood aggregation, WL encodes higher-order structural context around each node. Based on this, nodes that share identical or similar WL signatures can be interpreted as occupying identical or similar structural roles within the graph, even if their features may differ.
>
> 3. We further expanded Section 3.2 (pages 5-6) to explain why WL was selected as a structural metric. Our motivation is not simply that WL empirically performs well, but that it provides a principled and computationally efficient way for capturing higher-order neighborhood structure. In contrast to simpler metrics such as degree centrality or local neighborhood overlap, WL progressively captures multi-hop structural information and can distinguish nodes that share similar local statistics even if these nodes are far from each other in the graph.
>
> 4. Regarding Figures 1-2, we would like to clarify that the one depicting the WL algorithm (Figure 2 in the first version of the manuscript) is intended only for illustrative purposes and that in the figure, nodes that receive the same color after the 1st iteration of the WL algorithm (i.e., nodes that have the same degree) are depicted as similar to help understand the proposed method. However, in our actual methodology (Figure 1 in the first version of the manuscript), structural similarity is not determined from the colors produced at a single WL iteration alone. Instead, we construct the WL tree induced by the sequence of WL refinements and measure structural similarity through the shortest-path distance between the corresponding leaf nodes. Thus, the final WL colors identify the leaves of the tree, while the resulting distance reflects the shared refinement history of two nodes within the WL hierarchy. We believe the reviewer’s concern may have arisen from the original ordering of the figures, in which the illustration of our methodology preceded the figure describing the WL algorithm. To avoid any possible confusion, we have revised the manuscript (Section 3 at page 4) and swapped the positions of the two figures.

---

### Review · Reviewer_j2vM · 2026-06-06

**Summary Of Contributions:**

# Summary
The paper introduces WLGCL, a graph contrastive learning framework that combines similarity in the learned feature space with structural similarity derived from Weisfeiler–Leman refinement. For each anchor node, the method independently retrieves feature-similar and structure-similar candidates, treats their intersection as additional positive samples, and uses nodes selected by only one view as hard negatives. These samples are incorporated into a multi-positive contrastive objective augmented with MoCHi-style interpolated negatives. The method is evaluated on six homophilic and heterophilic node-classification datasets against seven graph self-supervised learning baselines, where it reports consistent improvements, particularly on heterophilic graphs. The paper also provides ablations on the encoder, loss design, similarity space, WL-tree depth, and computational efficiency.

## Strengths
* The paper addresses a relevant limitation of feature-driven contrastive sampling by explicitly incorporating graph structure.
* The dual-view sampling strategy is intuitive, interpretable, and relatively easy to integrate into existing graph contrastive learning pipelines.
* The emphasis on the shortcoming of the feature-based positive and negative minning for heterophilic graphs is very clearly stated and motivated for the proposal of the paper.
* The method reports consistent improvements across both homophilic and heterophilic datasets, with larger gains in the heterophilic setting.
* The experimental section includes useful ablations on the encoder, loss components, feature-similarity representation, and WL depth.
* The paper considers the trade-off between predictive performance and training time rather than reporting accuracy alone.

## Weakness
* The evaluation is limited to relatively small, transductive node-classification benchmarks, leaving the scalability and generality of the method unclear.
* The theoretical discussion of alignment and uniformity is mostly qualitative and does not formally establish the claimed benefits of the proposed sampling strategy.
* Treating all nodes that are similar in only one view as hard negatives is not sufficiently justified. In particular, structure-similar but feature-dissimilar nodes may be valid positives in heterophilic graphs. Additionally, it is discussed why among the non-selected k-hop neighbors of an anchor node the negative samples would not be chosen. The ignored subset of the k-hop neightbors by the two views may be viewed as negative samples.
* The stated O(∣V∣log∣V∣) complexity for feature-based retrieval appears inconsistent with computing cosine similarity between every anchor and all other nodes, unless an approximate nearest-neighbor procedure is used and described.
* The method introduces several dataset-specific hyperparameters, including the feature and structural thresholds, but the sensitivity and tuning cost of these parameters are not thoroughly evaluated.
* The fairness of the comparison requires further clarification because WLGCL uses the structurally expressive WLHN encoder and tunes thresholds separately for each dataset, while the baselines are reportedly run using their default hyperparameters and encoders.

**Audience:**

Yes

**Audience Explanation:**

The paper is very well motivated and instructed to help the audience follow the abstraction, formulation,and experimentation. The evaluation is made on top of reproducible setups and the baselines are broad and extensive for the analysis of the results. Therefore, we find it insightful and guiding the audience towards future works and novelties.

**Broader Impact Concerns:**

Not applicable. No significant ethical or broader impact concerns are apparent that would require additional discussion.

**Claims And Evidence:**

No

**Claims Explanation:**

Despite some gap between the claim and evidence, it seems the broad implication of the paper is such that the claims are supported by the evidence up to the following improvement and adjustment.

**Requested Changes:**

Change requests are ordered from the high to the low priority.

1. **Critical — Establish a fair and controlled baseline comparison.**
   The main comparisons should be repeated under a standardized protocol, including the same encoder architecture where applicable, comparable hyperparameter-search budgets, identical data splits, and consistent training and evaluation procedures. WLGCL uses the WLHN encoder and dataset-specific threshold tuning, whereas the baselines are reportedly evaluated using their default hyperparameters. This makes it difficult to attribute the reported improvements specifically to the proposed sampling strategy. The experimental settings of the baselines and whether they are re-generated or reported by the other (original) paper are not explained.

2. **Critical — Provide a complete component-wise ablation study.**
   The authors should separately evaluate the contributions of:

   * WL-based structural candidate retrieval;
   * intersection-based positive extension;
   * feature-only hard negatives;
   * structure-only hard negatives;
   * cross-view inconsistent hard negatives;
   * MoCHi-style harder and hardest negative interpolation.

   The current comparison between InfoNCE, MoCHi, and the full WL-Contrast objective changes several components simultaneously and therefore does not clearly identify which components are responsible for the observed improvements.

3. **Critical — Justify treating single-view-similar nodes as hard negatives.**
   The method assumes that nodes similar in only one of the feature or structural views should be treated as negatives. This may introduce false negatives, particularly in heterophilic graphs, where structurally similar nodes may legitimately have dissimilar features. The authors should quantify the semantic or label consistency of the selected positives and negatives, analyze how often the proposed rule produces false negatives, and compare against alternatives in which uncertain samples are ignored or weighted rather than explicitly pushed apart.

4. **Critical — Correct or substantiate the computational complexity claims.**
   The stated per-epoch complexity of (O(|V|\log |V|)) for feature-similarity computation is not clearly consistent with computing similarities between every anchor node and all other nodes. The paper should provide a precise complexity derivation, explain whether approximate nearest-neighbor retrieval or another indexing mechanism is used, and report memory consumption and runtime scaling as graph size increases.

5. **Critical — Support or narrow the broader claims.**
   Claims regarding scalability, robustness to feature noise, robustness under distribution shift, encoder agnosticism, and general plug-and-play applicability are not directly established by the current experiments. The authors should either provide targeted experiments supporting these claims or revise the corresponding statements so that the conclusions are restricted to the evaluated transductive node-classification setting.

6. **Important — Evaluate the method on larger and more diverse graph settings.**
   The experimental evaluation is limited to six relatively small node-classification benchmarks. Including substantially larger graphs, inductive learning settings, or additional downstream tasks such as link prediction would provide stronger evidence that the method generalizes beyond small transductive benchmarks. Otherwise, the claim needs to be limited to the node-level small-scale setups.

7. **Important — Analyze the sensitivity and tuning cost of the similarity thresholds.**
   The feature and structural thresholds, (\theta_{\mathrm{feat}}) and (\theta_{\mathrm{struct}}), are tuned separately for each dataset. The paper should report the search ranges, validation protocol, selected values, sensitivity curves, and computational cost of this tuning process. It would also be useful to compare absolute thresholds with percentile-based or top-(k) candidate selection.

8. **Important — Quantify the positive and negative candidate sets.**
   The authors should report statistics such as the average and distribution of ( |P_v^{\mathrm{feat}}| ), ( |P_v^{\mathrm{struct}}| ), ( |P_v^{\mathrm{inter}}| ), and ( |N_v^{\mathrm{hard}}| ) across datasets and, where relevant, across training epochs. This analysis would clarify the strength of the resulting contrastive signal and how the sampling behavior differs between homophilic and heterophilic graphs.

9. **Important — Clarify the contribution of the WLHN encoder.**
   Because WLHN already introduces a strong structural inductive bias and substantially outperforms the alternative encoders in the reported ablation, the authors should evaluate competing methods with the same WLHN backbone where technically possible. This would better separate the benefit of the proposed dual-view sampling strategy from the benefit of the encoder itself. Additionally, it is not clear in the case WLHN is the encoder, how much the structure-aware view contribute to the robustness gain on the node classification downstream task.

10. **Strengthening — Improve reproducibility and methodological clarity.**
    The paper should provide complete implementation details for graph augmentations, optimization, negative interpolation, batch construction, threshold selection, early stopping, downstream logistic-regression evaluation, and dataset splits. The apparent inconsistency between describing the proposed module as leaving the contrastive objective unchanged and subsequently introducing the new WL-Contrast objective should also be resolved.

12. **Strengthening — Revise unclear notation and presentation.**
    Several definitions should be clarified or corrected. In particular, the paper should clearly distinguish graph-neighborhood positives from augmentation-based positives, provide an unambiguous definition of the negative set, and precisely define the construction of the “hard,” “harder,” and “hardest” negative samples.

13. **Alignment-Uniformity — Provide missing references.**
References to the "the alignment-uniformity trade-off" concept  in the introduction is missing. It is not clear how the current method step towards a balanced trade-off.

---

> ### Author Response · Authors · 2026-06-26
>
> We thank the reviewer for their thoughtful evaluation and the requested changes, which we have incorporated in the revised version of the paper. Below, we provide detailed responses to the reviewer’s comments.
>
> **Answer: (Q1, Q9)**
> We have clarified the experimental protocol in the revised paper. All baseline methods were evaluated using their official code implementations and recommended hyperparameter settings, which are often dataset-specific in the original papers. To further verify that the performance gains do not simply originate from the WLHN encoder, we additionally included a standard GCL variant using the same WLHN encoder in the comparison (denoted as GCL+WLHN). As shown in Table 3 (page 11), WLGCL consistently outperforms this WLHN-based baseline, indicating that the observed improvements are primarily attributable to the proposed sampling and contrastive learning strategy rather than the encoder itself.
>
> **Answer: (Q2, Q3)**
> In the revised manuscript, we substantially expanded the ablation study in Section 4.3 (Tables 5 and 6, pages 12-13-14). Specifically, we separately evaluate feature-only, structure-only, and intersection-based positive expansion strategies, as well as different hard-negative construction mechanisms and contrastive loss components. The results show that the proposed intersection positives consistently achieve the highest label agreement across all datasets, confirming that they form a more reliable positive set. We further analyze the proposed hard negatives and find that the dual-view hard-negative set exhibits intermediate agreement levels between positives and random negatives, indicating that it provides a challenging yet informative contrastive signal rather than introducing large numbers of trivial false negatives. While some of these samples may share labels with the anchor, their consistency is substantially lower than that of the intersection positives, suggesting that they occupy an ambiguous region between reliable positives and trivial negatives. Therefore, we treat these cross-view inconsistent samples as informative hard negatives rather than positive extensions. We agree that alternative treatments such as ignoring or softly weighting uncertain samples are interesting directions and will investigate them in future work.
>
> **Answer: (Q4)**
> In the revised manuscript, we corrected and expanded the complexity analysis in Section 3.6 (page 9), providing a more rigorous derivation of both the worst-case time and space complexities of the proposed method. We also discuss the primary computational bottlenecks and potential directions for further improving scalability through more efficient retrieval and indexing strategies.
>
> **Answer: (Q5)**
> We have revised the manuscript to narrow the claims regarding robustness to feature noise and distribution shift. Regarding encoder agnosticism and plug-and-play applicability, we believe these claims remain supported by both the formulation of our method, which operates independently of a specific graph encoder, and the experimental results demonstrating consistent improvements when combined with different contrastive learning frameworks.
>
> **Answer: (Q6)**
> We agree that evaluating on larger-scale graphs and additional downstream tasks would provide stronger evidence of generalization. We would like to note that the six datasets used in our experiments are standard and widely adopted benchmark datasets in the graph contrastive learning literature, and were selected to facilitate fair comparison with prior work. Due to the memory limitations of our available hardware and the limited revision period, we were unable to conduct additional experiments on substantially larger datasets. Consequently, we restrict our empirical conclusions to the evaluated node classification benchmarks. At the same time, the proposed sampling framework is not inherently restricted to small graphs, and the revised complexity analysis discusses how the method could be extended to larger-scale settings through more efficient implementations.
>
> **Answer: (Q7)**
> We have added a detailed description of the hyperparameter tuning procedure in Appendix A (page 19), including the search ranges, validation protocol, and the selected values for different datasets.
>
> **Answer: (Q8)**
> We have added statistics of the sampled candidate sets in Appendix B (page 19), including the sizes of the feature-based, structure-based, intersection-based positive sets, as well as the hard-negative sets across different datasets.

---

> > ### Author Response · Authors · 2026-06-26
> >
> > **Answer: (Q10)**
> > We will soon release the official implementation together with the complete experimental configurations and hyperparameter settings used in our experiments. These materials will include details regarding optimization, augmentation, threshold selection, training procedures, and evaluation settings. Regarding the apparent inconsistency in the presentation, we would like to clarify that our method does not alter the fundamental objective of contrastive learning, which remains to pull positive samples closer and push negative samples apart. The proposed WL-Contrast objective modifies the construction of positive and negative samples while preserving this underlying contrastive principle.
> >
> > **Answer: (Q11)**
> > Regarding the definitions of positive and negative samples, we believe these components are already described in detail in Sections 3.4 and 3.5. Specifically, the construction of graph-neighborhood positives, expanded positives, hard negatives, harder negatives, and hardest negatives is formally defined, and the corresponding procedures are accompanied by references to the underlying methods on which they are based. We have revisited these sections to ensure that the notation and definitions are presented consistently throughout the manuscript. We hope the current formulation provides sufficient clarity regarding the construction of the different sample sets.
> >
> > **Answer: (Q12)**
> > In the Introduction, and as suggested by the reviewer, we provided further references to the alignment-uniformity trade-off (page 2). In this respect, we also included additional sampling statistics in Appendix B, which give further insight into the semantic consistency of the constructed positive and negative sets. Although we do not directly optimize or explicitly measure alignment and uniformity, these observations are consistent with the intuition that WLGCL promotes a better balance between the two objectives.